# Mapping Resistance to Argentinean *Fusarium* (*Graminearum*) Head Blight Isolates in Wheat

**DOI:** 10.3390/ijms222413653

**Published:** 2021-12-20

**Authors:** Carolina Sgarbi, Ismael Malbrán, Luciana Saldúa, Gladys Albina Lori, Ulrike Lohwasser, Mian Abdur Rehman Arif, Andreas Börner, Marcos Yanniccari, Ana Maria Castro

**Affiliations:** 1Biological Science Department, UNNOBA, Buenos Aires B6000, Argentina; csgarbi@comunidad.unnoba.edu.ar; 2National Council for Scientific and Technological Research (CONICET), La Plata 1900, Argentina; ismael.malbran@gmail.com (I.M.); marcosyanniccari@conicet.gov.ar (M.Y.); castroam@gmail.com (A.M.C.); 3Centro de Investigaciones en Fitopatología (CIDEFI-UNLP-CIC), La Plata 1900, Argentina; gladyslori@hotmail.com; 4Genetics, Faculty of Agricultural Sciences, UNLP, La Plata 1900, Argentina; lcsaldua@gmail.com; 5Leibniz-Institut für Pflanzengenetik und Kulturpflanzenforschung (IPK), D-06466 Seeland, Germany; lohwasse@ipk-gatersleben.de; 6Wheat Breeding Group, Plant Breeding and Genetics Division, Nuclear Institute for Agriculture and Biology, Faisalabad 38000, Pakistan; 7Chacra Experimental Integrada Barrow (MDA-INTA), Tres Arroyos B7500, Argentina; 8Centro de Investigación en Sanidad Vegetal (CISaV), FACAyF, UNLP, La Plata 1900, Argentina

**Keywords:** wheat, FHB, QTL mapping, ITMI, epistasis

## Abstract

*Fusarium* head blight (FHB) of wheat, caused by *Fusarium graminearum* (Schwabe), is a destructive disease worldwide, reducing wheat yield and quality. To accelerate the improvement of scab tolerance in wheat, we assessed the International Triticeae Mapping Initiative mapping population (ITMI/MP) for Type I and II resistance against a wide population of Argentinean isolates of *F. graminearum.* We discovered a total of 27 additive QTLs on ten different (2A, 2D, 3B, 3D, 4B, 4D, 5A, 5B, 5D and 6D) wheat chromosomes for Type I and Type II resistances explaining a maximum of 15.99% variation. Another four and two QTLs for thousand kernel weight in control and for Type II resistance, respectively, involved five different chromosomes (1B, 2D, 6A, 6D and 7D). Furthermore, three, three and five QTLs for kernel weight per spike in control, for Type I resistance and for Type II resistance, correspondingly, involved ten chromosomes (2A, 2D, 3B, 4A, 5A, 5B, 6B, 7A, 7B, 7D). We were also able to detect five and two epistasis pairs of QTLs for Type I and Type II resistance, respectively, in addition to additive QTLs that evidenced that FHB resistance in wheat is controlled by a complex network of additive and epistasis QTLs.

## 1. Introduction

*Fusarium* head blight (FHB) or scab, caused mainly by *Fusarium graminearum* (Schwabe) and *F. culmorum* (WG Smith) Sacc, is one of the most important fungal diseases affecting wheat in cereal producing areas of the world [1,2,3]. The economic losses caused by FHB include yield and quality reduction. The damages induced by the disease are further aggravated by the frequent presence of mycotoxins in affected grains. These persistent, thermo-stabile metabolites, produced in association with food and feeds, may cause health problems to human and animals even in low doses [4,5]. At present, FHB is widely diffused in several Eastern European countries, Asia (China and Japan), North Africa, North America (USA and Canada) and South America, especially in Argentina, Brazil, Paraguay, and Uruguay [4].

In the recent years, monoculture, reduced tillage and maize/wheat rotations have greatly increased the level of inoculum in the soil and, hence, the risk for epidemics of FHB [5]. Although they can reduce the damage induced by the disease to some extent, agronomic practices and fungicides currently available are far from effective at preventing the occurrence of epidemics. Hence, the development of resistant cultivars is the most economic, effective and environmental friendly approach to manage this disease [6].

In this regard, significant progress has been achieved in wheat research which has culminated into the release of some resistant varieties [1,6,7,8,9]. However, the scarce understanding of the underlying genetic mechanisms of the resistance to FHB and the consequent lack of novel sources of resistance to the disease limit the potential progress in the development of resistant cultivars.

The resistance to FHB includes passive and active mechanisms. Passive mechanisms are associated with morphological traits such as height, awnedness and width of flower opening during anthesis [10]. Active mechanisms comprise the following types of resistance: (I) resistance to initial infection [11]; (II) resistance to spreading [11]; (III) resistance to kernel infection [10,12]; (IV) tolerance to infection [10,12]; and (V) resistance to deooxynivalenol (DON) mycotoxin accumulation [13].

Wheat resistance to FHB is inherited as a quantitative trait governed by polygenes, and quantitative trait loci (QTLs) have been detected on all wheat chromosomes [6,14,15,16,17,18]. Each locus had low contribution to heritability and was sensitive to genetic background [19,20,21,22,23]. To date, 200 QTLs for FHB resistance have been reported; however, most of these studies were focused on a few resistant cultivars, such as Sumai 3 and its derivatives [19,24,25,26,27,28,29,30,31,32]. Not long ago, another 14 QTLs were identified in two mapping populations from Brazil [33] where at least two QTLs were common across different environments. The investigators also located a major, novel QTL for DON accumulation on chromosome 4B and a major QTL associated with thousand grain weight on chromosome 6B. Search for resistance in natural populations has also been initiated using the associated mapping strategy [14].

Chinese germplasm, including the cultivars Sumai 3 and its Ning derivative, confers Type II resistance and has been extensively used in breeding programs worldwide to improve the levels of resistance to FHB [34]. A major QTL for FHB resistance is located on the short arm of chromosome 3B of Sumai 3 and its descendants (*Fhb-*1) [29,30,31,32,35,36] while additional QTLs were detected on chromosomes 5AS [35,37] and 6BS [38]. Another QTL associated with FHB resistance has been identified on chromosome 3A of the Brazilian resistant var. Frontana [39]. A research comprising 358 European winter wheat and 14 spring wheat varieties tested in four environments found associations with FHB resistance on all wheat chromosomes, except chromosome 6B [15]. However, these lines have provided only partial resistance.

The identification of these QTLs has assisted in clarifying the inheritance of the resistance to FHB in wheat [40]. Furthermore, QTL mapping and marker-assisted selection (MAS) could largely enhance the efficiency of novel germplasm use and aid in breeding resistant cultivars [6].

A strong association between FHB severity and DON concentration in infected grain has been reported [41,42]. Therefore, selection for lower FHB symptoms in segregating generations might lead to lower DON accumulation in the grain [42]. Nonetheless, other researchers failed to detect any association between FHB severity and DON concentration and suggested the possibility of independent QTLs/genes for resistance to DON accumulation and kernel infection from Type II resistance [21,25]. In this regard, a major QTL was identified on chromosome 2AS providing low DON content, but not low FHB severity [43]. It is expected that the accumulation of QTLs of these types of resistance will necessarily increase awareness of mycotoxin contamination and the damage to grain yield and quality induced by FHB [43]. To this point, the identification of effective resistance genes on different sources for heterogeneous populations of *F. graminearum* is critical to accelerate the improvement of scab resistance in wheat. The current research was aimed at searching new sources of FHB resistance for Types I and II against a wide collection of *F. graminearum* isolates obtained from the main wheat cropping area of Argentina.

## 2. Results

### 2.1. Phenotypic Variation

The different traits studied showed significant differences between genotypes (except for the FDK), treatments (except for the number of spikes, NS), and in the interaction (except for the FDK) (Table 1) during the three years. Parental lines showed highly significant differences in SE for both types of resistance. Furthermore, the Synthetic parent showed no significant differences with Sumai 3 for SE (Figure 1). The RILs’ mean value was similar to that of the Synthetic parent for both types of resistance. There were 66 RILs that showed lower than or similar SE as the resistant parent Synthetic when Type I resistance was evaluated. Similarly, 59 RILs showed lower or comparable SE to Synthetic when the Type II resistance mechanism was assessed (Figure 1).

Parental lines showed highly significant differences in SE for both types of resistance. Furthermore, the Synthetic parent showed no significant differences with Sumai 3 for SE (Figure 1). The RILs’ mean value was similar to that of the Synthetic parent for both types of resistance. There were 66 RILs that showed lower than or similar SE as the resistant parent Synthetic when Type I resistance was evaluated. Similarly, 59 RILs showed lower or comparable SE to Synthetic when the Type II resistance mechanism was assessed (Figure 1a).

FI parental lines showed highly significant differences for both types of resistance (Figure 2). The Synthetic parent showed no significant differences with Sumai 3 for either type of resistance in the case of FI. The RILs’ mean value was similar to that of the Synthetic parent. There were 91 and 90 RILs that showed similar FI to the resistant parent Synthetic for Type I or II resistance, respectively (Figure 1b).

Parental lines showed significant differences for TKW in the inoculated plants and between the inoculated and control plants of the Opata parent (Figure 1c). The RILs showed similar TKW mean values in the control and inoculated plants. Moreover, 23 inoculated RILs showed a TKW that exceeded the RILs’ mean value as well as the values of this trait recorded in parental lines. No significant differences in TKW were found between the inoculated and control plants of the FHB tolerant cultivar Sumai 3. Nonetheless, TKW of Sumai 3 was significantly lower than that recorded in Synthetic and 61 inoculated RILs (Figure 1c).

No significant differences were found between the FDK recorded in the Synthetic parent and in Sumai 3, neither for controls nor for Type I inoculated spikes (Figure 1d). On the other hand, highly significant differences were found between the FDK in the control and inoculated plants of the Opata parent. The RILs’ mean values for FDK were similar for both types of resistance (Figure 1d).

The number of spikes per plant was significantly different between RILs, and slightly different in the interaction (Table 1). The controls and inoculated plants showed similar NS for Synthetic, the RILs’ mean values and Sumai 3 (Figure 1e). Opata showed significantly higher NS than Sumai 3, Synthetic and the RILs’ mean values. There were 50 RILs with similar NS as Opata for Type I and 39 for Type II resistance (Figure 1e). The kernel weight per plant was significantly different between control and inoculated plants for both Opata and the RILs’ mean values (Figure 1f). There were no significant differences in the KWP for Synthetic and Sumai 3 either in control or inoculated plants.

There were no significant differences in the kernel weight per spike between Synthetic, Sumai3 and the RILs; mean values, or between their control and inoculated plants (Figure 1g). Opata showed significantly lower KWS when inoculated. There were 22 and 19 RILs with significantly higher KWS than the Synthetic parent for Type I and II resistance, respectively (Figure 1g).

### 2.2. QTL Analysis

The traits studied showed continuous segregations. Therefore, the QTL mapping technique was followed to detect the loci determining Type I or II resistance. The SE, FI, TKW and KWS for Type I and II resistance and the FDK for Type I resistance showed significant associations to molecular markers.

#### 2.2.1. Additive QTLs

A total of seven QTLs (five in 2008 and two in 2009) were uncovered regarding FI for Type I resistance on chromosomes 3D (two QTLs), 4B, 5A, 5B (two QTLs) and 5D (Table 2) (Figure 2). The LOD value for these QTLs ranged between 2.70 (on chromosome 5D) to 9.14 (on chromosome 3D). The PVE explained was 7.95–47.74%. On the other hand, eight QTLs were detected for Type II resistance of FI (two in 2008 and six in 2009) which were located on chromosomes 2A, 3B, 3D, 4B (two QTLs), 5A, 5D and 6B. The minimum and maximum LOD in this case was 2.02 and 4.80. These QTLs were responsible for 5.27 to 15.99% variation.

D-genome provided the four QTLs for SE of Type I resistance on chromosomes 2D, 4D and 5D (two QTLs), whereas another QTL was located on chromosome 4B. The LOD for these QTLs ranged from 2.12–4.51 whereas the PVE ranged from 6.49–15.09%. D-genome also provided the four QTLs for SE of Type II resistance on chromosomes 5D (three QTLs) and 6D. Two further QTLs were located on chromosomes 2A and 4B. The LOD for these QTLs was below 3 (2.06–2.36) and the PVE ranged from 3.74–10.09%. All of them were detected in 2009. For FDK Type I, there was only one QTL detected on chromosome 2B with an LOD of 2.79 that explained 13.63% variation, whereas no QTL could be detected for FDK Type II resistance.

Four QTLs (one in 2008 and three in 2010) explained the TKW in control plants on chromosomes 1B, 2D, 6D and 7D where LOD ranged between 2.48–3.60 and the PVE ranged between 16.27–19.90%. Chromosome 7D also carried related to TKW for Type II resistance (LOD: 2.03 and PVE = 14.90%) which was embedded in between the two QTLs of TKW of control plants. There was another QTL on 6A related to TKW for Type II resistance in 2009 with an LOD of 2.27 and 13.60% PVE.

KWS in control was controlled by three QTLs on chromosomes 3B (2010), 5A and 7A (2008). The LOD for these QTLs ranged between 2.03–2.24 and the PVE was 13.9–19.28. We could identify three QTLs related to KWS for Type I resistance in 2008 located on chromosomes 4A, 5A and 7D at LOD ranging between 2.15–2.50 responsible for 18.51–20.95% variation. Four QTLs controlled KWS for Type II resistance in 2010 on chromosomes 2A, 5B, 6B and 7B where the LOD ranged between 2.07–2.68 and the PVE was 14.21–20.19%. On the other hand, in 2008, another QTL on chromosome 2D was discovered related to KWS for Type II resistance at LOD of 3.04 responsible for 35% variation.

#### 2.2.2. Epistasis QTLs

In addition to seven additive QTLs, FI Type I resistance was also under the influence of six pairs of epistasis QTLs which were located on chromosomes 2B-2B, 2A-5B, 1B-5B, 5A-5B, 5A-5D and 5B-6A that were responsible for an additional 5.24–12.31% PVE, individually (Figure 2) (Table 3). SE Type I resistance was also under one pair of epistasis QTL on chromosomes 2B-3B, apart from the five additive QTLs. For FI Type II resistance, we uncovered two pairs located on chromosomes 5D-5D and 5D-6D where the PVE was 6.55–6.76%. Finally, another five pairs of epistasis QTLs were detected on chromosomes 2B-2B, 2D-2D, 2D-3B, 2A-6A and 1D-7D responsible for 5.82–10.34% variation for FDK Type I resistance.

## 3. Discussion

The development of resistant cultivars is the most economic, effective and environmental friendly approach to manage FHB [8]. The first step, however, is to identify the resistant sources in the targeted environments to formulate the future strategy to combat or manage that stress [44]. The ITMI/MP has been the most explored population for various genetic mapping studies including many different agronomic traits [45] and seed longevity and dormancy [46,47]. The extent of variation and the number of QTLs regarding FHB resistance has, however, not been explored exhaustively.

We discovered seven, five and one QTLs, correspondingly, related to FI, SE and FDK for Type I resistance in the ITMI/MP (chromosomes 2B, 2D, 3D (two QTLs), 4B (two QTLs), 4D, 5A, 5B (two QTLs) and 5D (three QTLs including one common QTL for FI and SE). Chromosomes 2B, 2D and 4D are known to carry QTLs related to FHB resistance at 64, 76 and 12 cM, respectively [18]. Chromosome 2D is also known to be resident of a resistance QTL in a soft European winter wheat collection [48] along-side chromosome 1B, 1D and 2D. The reported SNPs here on chromosome 2D (*CAP12_c1503_76* and *D_GBUVHFX02GV41H_67*) carried a QTL cluster, C-2D.8 carrying multiple QTLs of agronomic traits including yield and harvest index (HI), heading (Hd) and flowering time Flt), seeds per spike, lodging resistance (Ld) and glume color ([45] which evidenced a gene rich region which can provide a valuable starting point for further gene search in future. In addition, chromosomes 4B, 5B and 5D have shown consistent associations for FHB in another study [14]. On the other hand, the SNPs involved here on chromosomes 3D (*BS00033229_51* and *D_contig00455_358*), 4B (*wsnp_Ex_c4148_7494801* and *Kukri_rep_c71670_163*), 5B (*BS00065390_51*, *AX-95145462*, *RAC875_c278_1801* and *Excalibur_c48387_58*) and 5D (*Kukri_c13045_302* and *IAAV6265*) were recently reported to be associated with HI, spike length, Flt, waxiness, powdery mildew (Pm) and leaf rust (Lr) resistance and plant height (Ht) [45].

The QTLs related to FI and SE for Type II resistance in ITMI/MP were 14 on eight different wheat chromosomes (chromosomes 2A (two QTLs), 3B, 3D, 4B (two QTLs), 5A, 5D (four QTLs including one common QTL for FI and SE), 6B and 6D). Previously, [32] identified a major QTL on chromosome 3BS in a mapping population derived from Sumai 3. They found five QTLs associated to resistance to FHB, two of them had major effects and the other three minor effects on resistance. The major QTL derived from Sumai 3 (*Qfhs.ndsu-3BS*, Syn *Fhb1*), was located on chromosome arm 3BS and linked to marker *Xbcd907*. Later, it was reported the finding of two microsatellite markers flanking the locus *Qfhs.ndsu-3BS*: *Xgwm493* and *Xgwm533* that explained 25 to 41% of the variation in resistance to FHB in two mapping populations [31]. Another major QTL on chromosome 6BL has been validated on other populations [49]. Hence, our QTLs could mirror those detected in the aforementioned studies. On chromosome 2A, [14] reported two associations at 104 cM in 161 diverse wheat lines whereas we detected two QTLs at 30 and 291 cM, indicating the importance of chromosome 2A in gene discovery related to FHB resistance in wheat. Our two QTLs on chromosome 4B were at a distance of 47 and 97 cM. On the other hand, [14] detected three associations at 6.78, 34.15 and 75.65 cM regarding FHB. Hence, our QTL at 47 cM and the association of [14] at 34.15 cM could be comparable. Another QTL on chromosome 3D was located at 249 cM whereas the MTA detected by the aforementioned study was at 143 cM. Reference [14] also detected one MTA on each of short (67 cM) and long arm (at 198 cM) of chromosome 5D. On the contrary, all of our 5D QTLs were on long arm (at 270, 317–319 and 352 cM), hence, a comparison is difficult. Likewise, the same study also detected two and one MTA, respectively, on chromosomes 6B and 6D. Likewise, we also detected one locus on each of chromosome 6B and 6D, albeit at different locations. On chromosome 6B, a Sumai 3 derived QTL [31] and another meta-QTL is described [18]. The authors of also detected four markers linked with FHB on chromosome 6D. When compared to the study by [45], the SNPs linked with QTLs on chromosome 2A (*RAC875_c24364_307* and *wsnp_Ex_rep_c66358_64543089*), 3D (*TA020105-1083* and *AX-94637066*), 4B (*wsnp_Ku_c12503_20174234* and *Kukri_c28022_54*), 5A (*wsnp_RFL_Contig4307_5006558* and *Kukri_c12738_882*) and 5D (*IACX3123* and *TA015368-0126*) were also reported to control kernel, awn and glume color, Lr, Pm, waxiness, spike density, Ld and many other traits.

There were 88 SNPs in totality that were involved with the 44 QTLs (Table 2) of various traits of FHB resistance, of which 76 were unique. Of these 76 SNPs, sequences of 75 SNPs were available from [45]. To get an apprehension of probable candidate genes in FHB resistance, we performed the blast analysis of the reported SNPs using the NCBI database (https://blast.ncbi.nlm.nih.gov/Blast.cgi?LINK_LOC=blasthome&PAGE_TYPE=BlastSearch&PROGRAM=blastx) (accessed on 15 December 2021). Among them, 40 SNPs provided a direct hit to a known sequence linked to some candidate gene, 23 SNPs were associated with either uncharacterized or hypothetical protein whereas 13 did not yield any hits (Appendix A).

On chromosome 2A, the candidate genes linked with SNPs of QTL involved with Sev_Type_II_2019 were *UDP-galactose transporter 1-like* and *ABC transporter C family member 2-like*. The former, also known as hUGT1, increased the lignin content and hardness of leaves and stems in tobacco [50], whereas ABC transporters are involved in cuticular lipid secretion [51]. We believe that both the aforesaid genes provided Type II resistance against FHB in ITMI/MP to some degree by providing hardness to the plant tissue which restricted the entry of the pathogen and caused little damage. Another SNP involved with FI_Type_II_2009 QTL on chromosome 3B was linked with *aldo-keto reductase (AKR) family 4 member C10-like* gene. AKRs (*KR18A1* gene) have been reported as promising agents for the control of *Fusarium* pathogens and detoxification of mycotoxins in plants and in food/feed products [52]. More recently, the role of AKRs has also been highlighted in salt, drought and abscisic acid stresses in other plant species [53]. Two more genes associated with FI_Type_II_2009 QTL on chromosome 3D were *EEF1A lysine methyltransferase 2-like* and *cell division cycle protein 123 homolog*. Both these genes come from very large gene families that play various roles in cellular metabolic functions.

Another probable candidate gene linked with an SNP of the QTL of KWS_Type_I_2008 on chromosome 4A was *cleavage and polyadenylation specificity factor subunit 3-I-like*. The SNPs on chromosome 4B provided hits with *MRP3*
*(m**ultidrug resistance protein 3)*, *glutamate receptor 3.1-like*, *probable serine acetyltransferase 2*, *probable receptor-like protein kinase*, *cysteine-rich receptor-like protein kinase 6* and *ABC transporter C family member 13-like isoform X1* genes. *MRP3* is reported to be among the cellular transport-associated transcripts previously shown to be DON-induced in a toxin-resistant wheat and to be linked to the DON tolerance and FHB resistance quantitative trait locus *Fhb1* [54]. Glutamate receptors are known to be part of numerous physiological and developmental processes [55], development of leaf pubescence and may contribute to the ability to respond to an attack from a pest or pathogen [56]. *Serine acetyltransferase* (SAT) is the rate-limiting enzyme in cysteine biosynthesis [57]. Furthermore, wheat *ABC transporter* has been reported to contributed to both grain formation and mycotoxin tolerance [58]. In addition, kinases are known to regulate cell growth and proliferation as well as triggering and regulation of immune responses [59]. On chromosome 4D, the only candidate gene identified was *inorganic phosphate transporter 1–2*-like for Sev_Type_I_2009 QTL. Phosphate transporters are involved in acquisition of inorganic phosphate (Pi) by plant roots and are located at the cytoplasmic membranes of epidermal cells and root hairs [60]. Wheat utilizes high amounts of Pi and mature grains are the major sink for Pi utilization and storage. The role of this candidate gene in plant defense mechanism against biotic stresses is, however, unclear.

The genes on chromosomes 5A were mainly involved in Type I resistance and included *asparagine synthetase*, *microtubule-associated protein 70–1* and *protein kinase G11A*. The bread wheat *asparagine synthetase* gene family is composed of five genes, viz. *TaASN1*, *TaASN2*, *TaASN3.1*, *TaASN3.2* and *TaASN4*. Among them, *TaASN1s* are located on group 5 chromosomes and expressed during mid-development of grains [61]. Their role in defense mechanisms is unclear. In addition, we also identified *auxin-responsive protein SAUR 36-like* as a candidate gene for Type I resistance on chromosome 5A. The auxin pathway is one of the few susceptibility-associated pathways during *Fusarium* infection [62].

On chromosome 5B, the four candidate genes probably involved in FHB resistance were carotenoid 9,10(9′,10′)-cleavage dioxygenase-like isoform X2, ATP-dependent zinc metalloprotease FTSH 5, protein PHOX1-like and protein SUPPRESSOR OF FRI 4-like. The former two were linked with Type I resistance and the latter were involved with Type II resistance. The carotenoid cleavage deoxygenases cleave the carotenoids and apocarotenoids are produced by their actions which are known to play various roles in the growth and development of plants [63]. On the other hand, zinc metalloprotease is recently reported to be involved in thermotolerance of wild wheat relative [64]. SUPPRESSOR OF FRI 4 delays flowering in Arabidopsis [65]. In addition, another gene linked to flowering time on chromosome 5D associated with Type I resistance was flowering time control protein FY-like isoform X1. Hence, a delayed or early flowering could be activated because of the infection of FHB. Four more candidate genes for FHB resistance on chromosome 5D were alpha-L-arabinofuranosidase 1-like isoform X1, aspartokinase 1, chloroplastic-like isoform X2, MAP kinase kinase and aspartyl protease family protein 2-like. Alpha-L-arabinofuranosidase is known to affect nutritional quality and processing quality of wheat grain by controlling the content of non-starch polysaccharide in grain cell wall [66]. Aspartokinase catalyzes the phosphorylation of aspartate, which is the first step in the biosynthesis of the other ‘aspartate family’ amino acids: methionine, lysine, and threonine [67] Likewise, MAP kinase kinase gene families are involved in many physiological processes [68] and the aspartyl protease have been reported in our previous report as one of the candidate genes of seed longevity in the same population. One of the probable candidate genes on chromosome 6B is transcriptional corepressor LEUNIG HOMOLOG-like. The involved SNP Ex_c17379_1431 had a significant effect on grain protein content, gluten content and alveograph strength elsewhere [69]. LEUNIG has a putative role in the gene regulations in a number of different physiological processes in Arabidopsis including disease resistance, DNA damage response, and cell signaling [70]. Others (probable WRKY transcription factor 19 isoform X1, disease resistance protein RGA5-like and serine/threonine-protein kinase AFC1-like) could have deeper roles in plant defense mechanisms. Lastly, the most important genes on group 7 chromosomes include probable leucine-rich repeat receptor-like protein, TOM1-like protein 6 isoform X2 and proline iminopeptidase-like. TOM1-like protein 2 (transporter of mugineic acid) is important in the maintenance of micronutrient homeostasis [71]. With respect to proline iminopeptidase-like protein (PIPs), it is found that PIPs are virulence factors of phytopathogenic bacteria and fungi [72].

Literature about epistasis interaction of FHB is scanty. The authors of [48] discovered two pairs of epistasis markers on chromosomes 3B-4B and 3B/4A-7B, although the genetic variance provided by the markers was only 4.5–5.6%. More recently, [14] detected five epistasis pairs on chromosomes 1A-4B, 2A-7A, 3B-7A, 5A-5D and 5A-6D in growth room experiments (69.09% PVE). In another experiment (greenhouse), another five pairs were detected on chromosomes 1A-2D, 2A-4B, 2A-6B, 2D-3B and 3D-4B responsible for 66.42% variation in entirety. Our results are hence in concurrence with that of [14] for two pairs on chromosomes 2D-3B and 5A-5D. In fact, the PVE in [14] and our QTL on chromosomes 5A-5D was also very similar (12.95% and 12.25%, respectively) although the locations of the markers involved vary. Nevertheless, we can speculate that these epistasis pairs could involve the same genes. Hence, our study is in line with [14] in providing evidence of the complexity of FHB resistance which is regulated by multiple loci.

TKW in this study was controlled by four QTLs in control on chromosomes 1B, 2D, 6D and 7D. This population has revealed 43 TKW QTLs on various chromosomes of bread wheat [45]. For TKW in inoculated plants, we discovered two QTLs on chromosome 6A and 7D. We do not have reports that discussed the effect of FHB on TKW. It can be stated, therefore, that these QTLs can be a breeding target to induce FHB resistance in the future wheat cultivars.

We also detected three minor QTLs for KWS on chromosomes 3B, 5A and 7A where 5A and 7A are known to carry multi-environmental QTLs for KWS by [45]. QTLs for KWS in inoculated plants were discovered to be located on chromosomes 2A, 2D, 4A, 5A, 5B, 6B, 7B and 7D, where only the QTL on chromosome 2D surpassed the highly significant threshold of LOD > 3.0 which overlaps with an SE Type I resistance QTL. This hints that the 2D QTL is involved in multiple processes in ITMI/MP in addition to providing resistance to FHB.

Elsewhere, it has been reported that the chromosomes 3B and 7A from Sumai 3 reduced DON accumulation within the kernels, while chromosomes 1B, 2D and 4D caused an increase of the trait [29]. The authors of [73] studied the effects of individual chromosome arms on FHB infection and DON accumulation using a set of ditelosomic lines derived from Chinese Spring. The authors of [74] suggested that chromosome arms 1DL, 2AL, 3AL, 1AL, 3BS and 1BS might carry genes contributing to resistance to DON accumulation, whereas on 7AS, 4DS, 6AS, and 6DL there might be susceptibility factors or resistance suppressors. Regardless of the results reported previously, we conclude that 4D and 6D carry favorable alleles provided by Synthetic, which improved both Type I and II resistances to FHB.

QTLs of minor magnitude have been identified on chromosomes 3BL, 3A and 5B [75], 4B [21,25], 4BL [32], 5A [30], 5DL [23,25], 6A and 6B [31]. Nonetheless, most of these QTLs were identified in Sumai 3 and its derivates. The genes reported here are novel since these resistance loci provided by Synthetic wheat on chromosomes 4DL, 5DS, 6DS and 7DL have not been described previously. Although the QTLs on 5D and 6D vary in their effects, the incorporation of these new QTLs/genes in wheat genotypes already carrying other loci might confer tolerance to FHB and result in gene pyramiding to control this pathogen.

## 4. Materials and Methods

### 4.1. Plant Materials

A set of 114 recombinant inbreed lines (RILs) commonly known as “International Triticeae Mapping Initiative mapping population” (ITMI/MP) derived from the cross of Opata and W7984 (a Synthetic wheat originated from the cross of *Triticum tauschii* and Altar 84) [76] and both parents were screened. The Chinese cultivar Sumai 3 was included in the trials as a resistant check.

Three sets composed of 20 plants of each RIL were sown individually in plastic pots (20 l) and cultivated under a shelter to avoid the rains. Two methods of inoculation were tested, aimed at identifying the effect of the two main sources of resistance to FHB currently identified. The aspersion of wheat spikes with a *F. graminearum* spore suspension was used to evaluate Type I resistance while the point inoculation (PI) of small volumes of a spore suspension inside some of the spikelets in the spike was used to evaluate Type II resistance [77]. One set of each genotype was inoculated by aspersion, one set by PI, and the last one remained uninoculated to serve as a control of natural infection. Trials were carried out from 2008 to 2010 in the field of the Experimental Station of the Facultad de Ciencias Agrarias y Forestales, UNLP, located at La Plata (34°55′ SL, 57°57′ WL), Argentina.

### 4.2. Source and Maintenance of F. graminearum Isolates

Forty-seven *F. graminearum* isolates were obtained from grain samples of common wheat (*Triticum aestivum* L.) collected from the main wheat cropping area of Argentina. Isolation of *F. graminearum* from infected grains, taxonomic identification and maintenance of the isolates were carried out as described in [78].

For inoculum production, isolates were cultured in Erlenmeyer flasks containing 50 mL of liquid carboxymethyl cellulose (CMC) medium [79] and incubated at 25 °C under agitation at 150 revs min^−1^ for 5 days. Macroconidia were harvested by centrifugation at 3500 revs min^−1^ at 2 °C and pelleted spores were resuspended in sterile distilled water. The spore concentration used for inoculation was obtained by mixing equivalent proportions of conidia from each of 42 *F. graminearum* isolates. The macroconidia concentration was adjusted to ≈10,000 spores mL^-1^ using a hemacytometer.

### 4.3. Inoculation Method

Wheat spikes were inoculated at anthesis (Zadoks growth stage 65) [80] with the *F. graminearum* conidial suspension. For Type I resistance evaluation, each spike was sprayed with 1 mL of the suspension using a manual atomizer (constant volume). For Type II resistance evaluation, 5 μL of the macroconidial suspension was pipetted between the lemma and palea of floret of each of the two middle spikelets of each spike. To keep ≈100% relative humidity and enhance spore germination, inoculated spikes were covered with plastic bags for 48 h.

### 4.4. Phenotypic Assessments

Spikes were visually rated for disease severity 21 days post inoculation (dpi). For both resistance types, FHB severity (SE) was calculated as the percentage of diseased spikelets over the total number of spikelets on each spike [81]. The total number of spikes on each treatment and the number of spikes with at least one symptomatic spikelet were also recorded and used to calculate FHB Incidence (I). The SE and I values obtained were used to calculate the Fusarium Index (FI: I/SE × 100).

At harvest the complete trial was hand threshed, and yield and quality parameters were recorded. Fusarium damaged kernels (FDK) Index was determined by visually inspecting for gray-white or pink discoloration of grains and mycelial growth. The thousand kernel weight (TKW) for each treatment was determined by standard methods and the number of spikes (NS), the kernel weight per plant (KWP) and the kernel weight per spike (KWS) were calculated. Data taken as percentage were arcsin-transformed prior to analysis. The data were analyzed by ANOVA using PROC GLM (SAS 1998), and the least significant difference (LSD) was used to test the differences between means. In order to present the variability found between RILs over three years the figures were drawn with the average for each variable.

### 4.5. Construction of Genetic Linkage Map and QTL Analysis

Details about genetic map construction are available in [45]. Briefly, DNA was extracted from young leaves and consequently used in genotyping employing the Illumina (San Diego, CA, USA) Infinium technology. An optimized array (wheat 20K Infinium SNP array) was used. This array is a refined version [82] of the 90K iSELECT SNP-chip described by [83]. To this chip, 5385 markers from the 35K Wheat Breeders Array [84] were also added. All the sequences of used SNPs are available in [45]. *Joinmap* (version 3.0) software [85] was used to construct the final genetic map which is available in [45].

To map the loci/genes linked with the FHB resistances, genotypic data consisting of 7845 high quality SNPs [45] were used. These SNPs were mapped to 92 of the original 114 RILs of the original population. For QTL analysis, inclusive composite interval mapping (ICIM) method embedded in *IciMapping* (version 4.2.53) was harnessed where the walking speed was kept 1.0 cM. An LOD score of >2 was applied to detect QTLs as significant and >3.0 as highly significant. Although many methods are available for genetic mapping in many other programs (QTL cartographer, Q gene, etc.), we preferred the ICIM because it has enhanced detection power and less biased estimates with reduced false detection rate [86]. Digenic epistatic QTLs were detected using the ICIM-EPI command where the LOD was kept 5.0 cM. Only the QTLs with LOD > 5.0 and explaining >5.0 phenotypic variation (PVE) were reported. QTLs were designated following the rules set out in the Catalogue of Gene Symbols [57,87]. All the additive and epistatic QTLs were visualized using the “*circlize*” package (version 0.4.13) in R [88].

## 5. Conclusions

All in all, this study produced a framework of multiple QTL/gene/s network of FHB resistance in ITMI/MP. New insights can be achieved by an understanding of the interactions between the QTLs and the environments where they are expressed. This will also shed light on the mechanisms in the background of FHB resistance. Prediction of single gene or multiple genes at the location of the reported QTLs at this time point cannot be stated. Hence, functional studies are needed to validate the actual role of reported SNPs in host-parasitism.

## Figures and Tables

**Figure 1 ijms-22-13653-f001:**
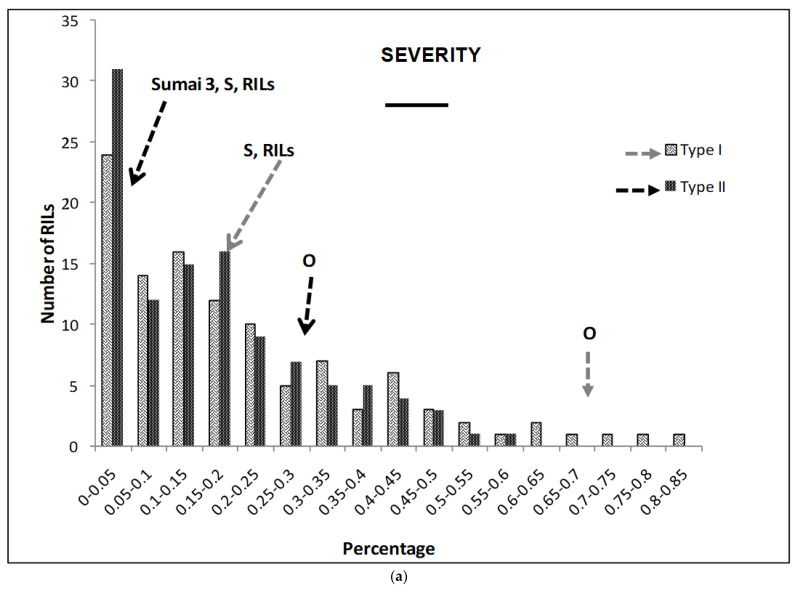
(**a**–**g**) Severity (**a**), Fusarium Index (**b**), thousand kernel weight (TKW) (**c**), Fusarium damaged kernel (FDK) (**d**), number of spikes (**e**), kernel weight per plant (**f**), and kernel weight per spike (**g**), recorded in ITMI/MP (Opata (O) × Synthetic (S)) assessed by Type I and II resistance and in their controls. Horizontal bar represents the standard error. Arrows indicate the mean values of parental lines, RILs and tolerant cultivar Sumai 3.

**Figure 2 ijms-22-13653-f002:**
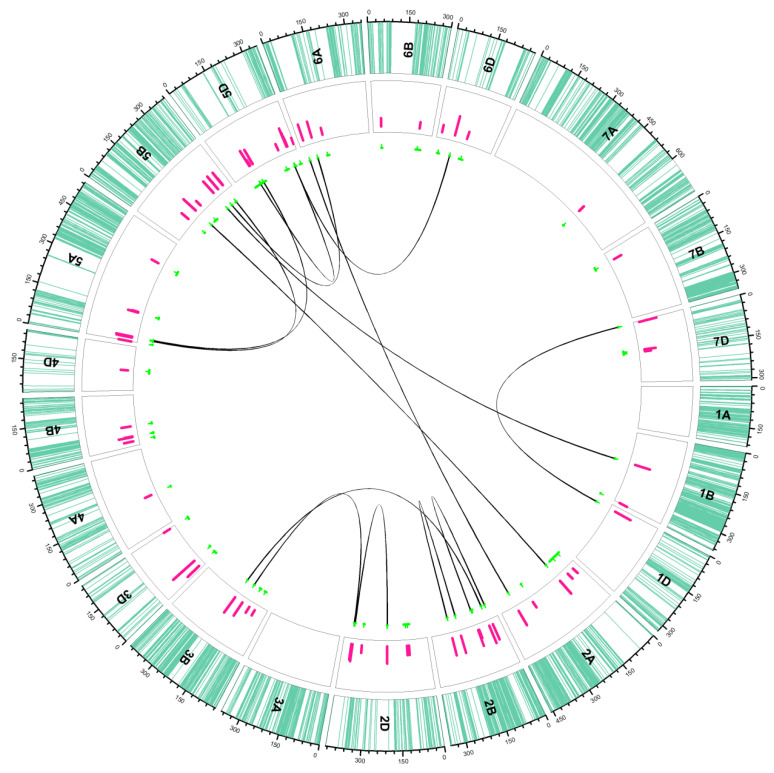
Distribution of additive (unconnected green lines in the inner circle) and epistatic (connected green lines in inner circle) QTLs in the ITMI/MP. Sea green lines in the outer track indicate the SNP positions on each chromosome; pink bars in the second circle indicate the LOD values of QTLs. The green lines under the track circle indicate the confidence interval of QTLs with small vertical lines point to the peak position of QTL. The black lines linked different biallelic epistasis QTLs. For details, see Table 2 and Table 3.

**Table 1 ijms-22-13653-t001:** ANOVAs for Severity, Fusarium Index, thousand kernel weight (TKW), Fusarium damaged kernels (FDG), number of spikes, kernel weight per plant and kernel weight per spike in ITMI/MP lines inoculated with *F. graminearum* for Type I and II of resistance and in the control plants. *, ** and *** indicate significant differences at *p*-values < 0.05, 0.01 and 0.001, respectively.

Sources	DF	Mean Squares						
		Severity	Fusarium Index	TKW	FDK	Number of Spikes	Kernel Weight per Plant	Kernel Weight per Spike
Genotypes	112	0.119 ***	0.054 ***	135.69 **	0.0078732ns	25.89 **	74.78 **	0.55 *
Treatment	2	1.476 ***	0.157 ***	920.91 ***	0.1735337 ***	35.68ns	535.02 ***	4.32 ***
G × T	224	0.101 ***	0.030 **	93.96 **	0.008631ns	12.57 *	46.80 **	0.45 **
Error	1029	0.000	0.005	73.09	0.0075382	12.09	32.81	0.44

**Table 2 ijms-22-13653-t002:** Traits for Type I and II resistance against FHB that were significantly associated to marker loci, chromosome linked to, the additive effects, the level of significance and the variability explained, determined in ITMI/MP inoculated with *F. graminearum.* Severity: Sev_Type_I and Sev_Type_II; *Fusarium* index: FI_Type I and FI_Type II; thousand kernel weight in control (TKW_C) and for type I (TWG_Type I); *Fusarium* damaged kernel: FDK I; kernel weight per spike in control (KWS_C) and for Type I (KWS_Type I) and Type II (KWS_Type II). Markers involved in multiple QTLs are in bold. *, **, ***, ^+^ indicate same QTLs and loci. Positive sign in the Add column indicates that the positive effect is imparted by “Synthetic” parent and the negative sign indicates the positive effect being provided by “Opata”.

Trait	Specific Trait	Chr.	Pos.	Left Marker	Right Marker	LOD	PVE (%)	Add	LeftCI	RightCI
**Type I resistance**	FDK_Type_I_2009	2B	114	*AX-94381628*	*BobWhite_c8253_397*	2.8	13.63	−4.13	111.5	116.5
	Sev_Type_I_2009 ^+^	2D	97	** *CAP12_c1503_76* **	** *D_GBUVHFX02GV41H_67* **	**3**	9.22	0.07	85.5	115.5
	FI_Type I_2008	3D	15	*BS00033229_51*	*D_contig00455_358*	**5.14**	16.61	−0.06	1.5	18.5
	FI_Type I_2008	3D	50	*D_GDS7LZN02IJRXZ_309*	*CAP12_c2615_128*	**9.14**	47.74	0.10	49.5	55.5
	FI_Type I_2008	4B	23	*wsnp_Ex_c4148_7494801*	*Kukri_rep_c71670_163*	**3.04**	7.84	−0.04	17.5	25.5
	Sev_Type_I_2009	4B	45	*tplb0027f12_503*	*BS00035426_51*	**4.51**	9.85	0.07	43.5	45.5
	Sev_Type_I_2009	4D	131.5	*AX-94838884*	*AX-95126745*	2.35	6.49	0.06	119.98	139.98
	FI_Type_I_2009	5A	0	*wsnp_Ex_c905_1748920*	*AX-95114232*	**4.2**	15.15	0.06	0	0.5
	FI_Type I_2008	5B	114	*BS00065390_51*	*AX-95145462*	2.88	8.03	−0.04	112.5	115.5
	FI_Type_I_2009	5B	356	*RAC875_c278_1801*	*Excalibur_c48387_58*	2.8	10.11	0.05	349.5	356.5
	Sev_Type_I_2009 *	5D	109	** *Kukri_c13045_302* **	** *IAAV6265* **	2.12	15.09	0.09	70.5	139.5
	FI_Type I_2008 *	**5D**	**124**	** *Kukri_c13045_302* **	** *IAAV6265* **	2.7	7.95	−0.04	87.5	137.5
	Sev_Type_I_2009	5D	318	** *AX-95190974* **	** *Kukri_rep_c106820_591* **	**3.35**	7.39	0.06	313.5	320.5
**Type II resistance**	FI_Type_II_2009	2A	291	*BobWhite_c4743_63*	*Excalibur_c47535_389*	2.16	5.57	0.032	288.5	291.5
	Sev_Type_II_2009	2A	30	*RAC875_c24364_307*	** *wsnp_Ex_rep_c66358_64543089* **	2.13	9.89	0.13	20.5	38.5
	FI_Type_II_2009	3B	84.2	*RAC875_c60169_200*	*BS00084607_51*	2.02	5.27	0.03	78.679	89.679
	FI_Type_II_2009	3D	249	*TA020105-1083*	*AX-94637066*	2.38	6.13	−0.03	238.5	252
	FI_Type_II_2009	4B	48	*Kukri_c32958_390*	*Excalibur_c19547_1012*	2.56	8.03	0.03	47.5	49.5
	Sev_Type_II_2009	4B	47	*wsnp_Ku_c12503_20174234*	*Kukri_c28022_54*	2.06	6.55	0.04	43.5	48.5
	FI_Type II_2008	4B	99	*AX-94899864*	*AX-94465680*	2.97	6.39	0.14	96.5	104.5
	FI_Type II_2008	5A	148	** *wsnp_RFL_Contig4307_5006558* **	** *Kukri_c12738_882* **	**3.35**	6.39	0.14	144.5	151.5
	Sev_Type_II_2009	5D	270	*IACX3123*	*TA015368-0126*	2.24	10.09	0.13	258.5	279.5
	FI_Type_II_2009 **	5D	317	** *AX-95190974* **	** *Kukri_rep_c106820_591* **	**4.8**	15.99	0.05	314.5	320.5
	Sev_Type_II_2009 **	5D	319	** *AX-95190974* **	** *Kukri_rep_c106820_591* **	2.25	3.74	0.05	312.5	323.5
	Sev_Type_II_2009	5D	352	*RAC875_c16419_585*	*BS00078603_51*	2.07	5.6	0.04	344.5	355
	FI_Type_II_2009	6B	233	*GENE-0221_350*	*Kukri_c32307_481*	2.06	5.36	−0.03	224.5	250.5
	Sev_Type_II_2009	6D	153	*AX-95107291*	*Excalibur_rep_c99143_422*	2.36	7.35	0.05	142.5	162.5
**TKW**	TKW_C_2010	1B	366	*AX-95154820*	*AX-94621372*	2.72	16.27	5.39	364.5	366
	TKW_C_2008	2D	332	*TA021271-0482*	*Excalibur_c1451_660*	2.48	18.04	−4.34	328.5	334.5
	TKW_Type II_2010	6A	126	*Excalibur_c23748_452*	*AX-94978875*	2.27	13.6	−6.03	122.5	134.5
	TKW_C_2010	6D	24	*AX-94633926*	*IACX10982*	2.36	19.9	6.17	19.5	31.5
	TKW_C_2010 ***	7D	145	*BS00066128_51*	** *Ku_c32426_324* **	**3.6**	19.9	−6.17	140.5	149.5
	TKW_Type II_2010	7D	155	*Kukri_c15768_1383*	*BS00062644_51*	2.03	14.9	−4.25	146.5	158.5
**KWS**	KWS_Type II_2010	2A	67	** *wsnp_Ex_rep_c66358_64543089* **	*Kukri_c33374_1048*	2.23	15.55	0.18	38.5	92.5
	KWS_Type II_2008 ^+^	2D	107	** *CAP12_c1503_76* **	** *D_GBUVHFX02GV41H_67* **	**3.04**	35	0.37	90.5	117.5
	KWS_C_2010	3B	115.2	*BS00065934_51*	*BobWhite_c22370_352*	2.24	19.18	31.0	111.67	121.67
	KWS_Type I_2008	4A	168	*RAC875_c25124_182*	*wsnp_Ex_c24474_33721784*	2.25	19.72	−0.20	166.5	170.5
	KWS_C_2008	5A	151	** *wsnp_RFL_Contig4307_5006558* **	** *Kukri_c12738_882* **	2.03	19.28	0.80	144.5	153.5
	KWS_Type I_2008	5A	416	*IAAV1179*	*AX-94706027*	2.5	20.95	0.21	403.5	423.5
	KWS_Type II_2010	5B	206	*Kukri_c52_225*	*wsnp_Ku_c3102_5810751*	2.07	14.21	−0.18	194.5	215.5
	KWS_Type II_2010	6B	40	*BS00068245_51*	*Ex_c17379_1431*	2.68	17.3	−0.19	38.5	43.5
	KWS_C_2008	7A	512	*AX-94582130*	*Tdurum_contig45618_1089*	2.15	13.98	0.40	509.5	514.5
	KWS_Type II_2010	7B	48	*BobWhite_c44404_312*	*Ex_c101666_634*	2.68	31.19	−0.30	38.5	54.5
	KWS_Type I_2008 ***	7D	150	** *Ku_c32426_324* **	*BS00022610_51*	2.15	18.5	0.19	143.5	154.5

**Table 3 ijms-22-13653-t003:** Pairs of epistasis QTLs detected in the ITMI/MP. Markers in bold are involved in multiple interactions.

Trait	Specific Trait	Chr1	Pos1	LeftMarker1	RightMarker1	Chr2	Pos2	LeftMarker2	RightMarker2	LOD	PVE (%)	Add1	Add2	AddbyAdd
**Type I resistance**	FI_Type_I_2009	1B	160	*AX-94503785*	*BobWhite_c48071_144*	5B	295	*BobWhite_c17845_132*	*AX-94579117*	5.04	6.15	0.0507	0.0454	0.0735
	FDK_Type_I_2009	1D	25	*Excalibur_c27873_266*	*RAC875_c51493_471*	7D	5	*AX-94741998*	*BobWhite_c40479_283*	5.73	10.34	5.1292	−7.0357	−6.5405
	FI_Type_I_2009	2A	120	*wsnp_Ex_rep_c66358_64543089*	*Kukri_c33374_1048*	5B	170	*Excalibur_rep_c67473_264*	*RAC875_c19099_308*	5.45	10.5	0.069	0.0477	0.0837
	FDK_Type_I_2009	2A	375	*wsnp_Ex_c14953_23104041*	*RAC875_rep_c69619_78*	6A	25	*BS00098857_51*	*Kukri_c14679_1082*	5.34	6.43	−0.0818	−1.9157	−5.325
	FI_Type_I_2009	2B	40	*RAC875_c98387_130*	*RAC875_c30797_179*	2B	120	*Kukri_c18459_2622*	*AX-94588421*	5.39	5.24	−0.0101	−0.012	−0.0714
	Sev_Type_I_2009	2B	60	*RAC875_c27650_216*	*BS00010988_51*	3B	201.2	*Excalibur_c9001_569*	*RAC875_c195_499*	6.04	10.83	0.0343	−0.0503	−0.121
	FDK_Type_I_2009	2B	210	*Tdurum_contig47202_1699*	*RAC875_c41476_217*	2B	255	*AX-94430027*	*RAC875_c22429_249*	5.74	8.96	−2.4307	3.8205	−12.4406
	FDK_Type_I_2009	2D	205	*Kukri_c33486_128*	*Excalibur_c24307_739*	2D	380	*AX-94602542*	*AX-95115363*	5.51	11.24	5.9524	4.5552	7.3597
	FDK_Type_I_2009	2D	385	*AX-94702227*	*BS00086534_51*	3B	156.2	*IACX971*	*Kukri_c35146_2094*	5.08	5.82	−0.2105	0.7398	5.3925
	FI_Type_I_2009	5A	20	** *AX-94694404* **	** *wsnp_Ex_c16551_25061517* **	5B	355	*RAC875_c278_1801*	*Excalibur_c48387_58*	5.11	12.31	0.0668	0.1015	0.0997
	FI_Type_I_2009	5A	25	** *AX-94694404* **	** *wsnp_Ex_c16551_25061517* **	5D	125	** *Kukri_c13045_302* **	** *IAAV6265* **	5.26	12.25	0.0327	0.0683	0.089
	FI_Type_I_2009	5B	325	*Tdurum_contig45588_730*	*Excalibur_c8168_226*	6A	70	*AX-95180013*	*BS00074992_51*	5.32	10.7	0.0559	0.0674	0.1193
**Type II resistance**	FI_Type_II_2009	5D	105	** *Kukri_c13045_302* **	** *IAAV6265* **	5D	315	** *AX-95190974* **	** *Kukri_rep_c106820_591* **	5.72	6.76	0.0475	0.0648	0.0577
	FI_Type_II_2009	5D	315	** *AX-95190974* **	** *Kukri_rep_c106820_591* **	6D	90	*TA001847-0566*	*D_GB5Y7FA02FHK0M_407*	6.31	6.56	0.0708	−0.0443	−0.0622

## Data Availability

All the genotypic and phenotypic data are available on request to the corresponding authors.

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
