# Peer review of "Mapping Resistance to Argentinean Fusarium (Graminearum) Head Blight Isolates in Wheat"

_ijms, 2021, doi:10.3390/ijms222413653_

Round 1

Reviewer 1 Report

Good example of experimental results with wide impact for breeding practice.  new information about new loci with focus to Fusarium resistance. Did you analyse content of mycotoxins too? Do you have some information about correlation bettwen resistance and mycotoxins content in your accessions? I know, that 21 days after inoculation is relatively short period. In article are all basic info about importance of FHB for yield and quality, well description of recent problem in crop rotation etc. Good and detail analyse of genetic background Manhattan figure including. I have any other comments.

Author Response

Response to Reviewer 1 comments

  • Good example of experimental results with wide impact for breeding practice.  new information about new loci with focus to Fusarium resistance.

Dear Reviewer 1, Thank you very much for your encouraging comments.

  • Did you analyse content of mycotoxins too?

Unfortunately, we did not analyze mycotoxin contents in ITMI/MP so far, but that can be done at some later stage.

  • Do you have some information about correlation between resistance and mycotoxins content in your accessions? I know that 21 days after inoculation is relatively short period

Since we did not measure mycotoxins, we are unable to predict about any possible correlation between resistance and mycotoxins in our population

  • In article are all basic info about importance of FHB for yield and quality, well description of recent problem in crop rotation etc. Good and detail analyse of genetic background Manhattan figure including.

Many thanks again for nice remarks

  • I have any other comments.

Do you mean “don’t have any comments?”

Reviewer 2 Report

  • Ln 123; Change ‘Pathogen’ into ‘Source and maintenance of F. graminearum isolates’
  • Ln 135; Change ‘Procedures’ into ‘inoculation method’
  • Ln 143; Change ‘Assessments’ into ‘Phenotypic Assessments’
  • Ln 160; Change ‘Genetic analysis’ into ‘Construction of genetic linkage map and QTL analysis ’
  • Combine Figures 1-7 into a single figure with visible font size.
  • Compare the QTLs with previously reported QTLs, I suggest drawing the comparison figure
  • Detail the physical distance of major QTLs and candidate genes

Author Response

Response to Reviewer 2 comments

  • Ln 123; Change ‘Pathogen’ into ‘Source and maintenance of  graminearum isolates’

DONE

  • Ln 135; Change ‘Procedures’ into ‘inoculation method’

DONE

  • Ln 143; Change ‘Assessments’ into ‘Phenotypic Assessments’

DONE

  • Ln 160; Change ‘Genetic analysis’ into ‘Construction of genetic linkage map and QTL analysis ’

DONE

  • Combine Figures 1-7 into a single figure with visible font size.

We have combined the figure and named it Fig 1(a-g) and the size is much larger with legible text for easy understanding.

  • Compare the QTLs with previously reported QTLs, I suggest drawing the comparison figure

We appreciate your remarks that we should compare the QTLs with previously reported QTLs with a suggestion of including a comparison figure. In this regard, it should be mentioned here that Buerstmayr et al. (2009)* provided a comprehensive review of 52 QTL studies up-to 2009. Post 2009, there have been conducted 101 QTL studies conducted comprising 52 publications with a total of 58 analysed mapping populations which have been summarized just last year (Buerstmayr 2020)** where a very detailed table is provided about all the QTLs discovered. To add to it, 12 publications using association panels and 37 publications reporting about validating, fine-mapping or cloning previously located QTL are available. It is also pertinent to mention that all those studies were conducted using different marker systems and populations derived from different parents and the location of same QTLs vary on different populations, hence, the positions of those QTLs one single figure is not possible whereas such information is already available in table form in afore told study (Buerstmayr et al. 2020)**.

  • Detail the physical distance of major QTLs and candidate genes

Table S1 has been provided indicating the physical position in megabases of the SNPs involved in the QTLs (where available). Moreover, candidate genes using the BLASTX analysis are also added. Further details and possible explanations of all the candidate genes involvement in Fusarium resistance is provided in the running text of the MS.

 *Buerstmayr, H., Ban, T., and Anderson, J. A. (2009). QTL mapping and marker‐assisted selection for Fusarium head blight resistance in wheat: a review. Plant breeding 128, 1-26.

**Buerstmayr, M., Steiner, B., and Buerstmayr, H. (2020). Breeding for Fusarium head blight resistance in wheat—Progress and challenges. Plant Breeding 139, 429-454.
